# Involvement of Mast Cells in the Pathology of COVID-19: Clinical and Laboratory Parallels

**DOI:** 10.3390/cells13080711

**Published:** 2024-04-19

**Authors:** Andrey V. Budnevsky, Sergey N. Avdeev, Djuro Kosanovic, Evgeniy S. Ovsyannikov, Inessa A. Savushkina, Nadezhda G. Alekseeva, Sofia N. Feigelman, Viktoria V. Shishkina, Andrey A. Filin, Dmitry I. Esaulenko, Inna M. Perveeva

**Affiliations:** 1Department of Faculty Therapy, Voronezh State Medical University Named after N.N. Burdenko, Studencheskaya Street 10, 394622 Voronezh, Russia; budnev@list.ru (A.V.B.); ovses@yandex.ru (E.S.O.); nadya.alekseva@mail.ru (N.G.A.); s.feygelman@gmail.com (S.N.F.); 4128069@gmail.com (V.V.S.); filinan@yandex.ru (A.A.F.); 2Department of Pulmonology, I.M. Sechenov First Moscow State Medical University (Sechenov University), Trubetskaya Street 8, 119991 Moscow, Russia; djurokos13@gmail.com; 3Research Institute of Experimental Biology and Medicine, Voronezh State Medical University Named after N.N. Burdenko, Moskovskiy Avenue, 185, 394066 Voronezh, Russia; desaulenko79@gmail.com; 4Voronezh Regional Clinical Hospital No. 1, Moskovskiy Avenue, 151, 394066 Voronezh, Russia; perveeva.inna@yandex.ru

**Keywords:** mast cells, COVID-19, chymase, tryptase, carboxypeptidase A3, lung damage

## Abstract

Recent studies suggested the potential role of mast cells (MCs) in the pathology of coronavirus disease 2019 (COVID-19). However, the precise description of the MCs’ activation and the engagement of their proteases is still missing. The objective of this study was to further reveal the importance of MCs and their proteases (chymase, tryptase, and carboxypeptidase A3 (CPA3)) in the development of lung damage in patients with COVID-19. This study included 55 patients who died from COVID-19 and 30 controls who died from external causes. A histological analysis of the lung parenchyma was carried out to assess the protease profiles and degranulation activity of MCs. In addition, we have analyzed the general blood test, coagulogram, and C-reactive protein. The content of tryptase-positive MCs (Try-MCs) in the lungs of patients with COVID-19 was higher than in controls, but their degranulation activity was lower. The indicators of chymase-positive MCs (Chy-MCs) were significantly lower than in the controls, while the content of CPA3-positive MCs (CPA3-MCs) and their degranulation activity were higher in patients with COVID-19. In addition, we have demonstrated the existence of correlations (positive/negative) between the content of Try-MCs, Chy-MCs, and CPA3-MCs at different states of their degranulation and presence (co-adjacent/single) and the levels of various immune cells (neutrophils, eosinophils, basophils, and monocytes) and other important markers (blood hemoglobin, activated partial thromboplastin time (aPTT), international normalized ratio (INR), and fibrinogen). Thus, the identified patterns suggest the numerous and diverse mechanisms of the participation of MCs and their proteases in the pathogenesis of COVID-19, and their impact on the inflammatory process and coagulation status. At the same time, the issue requires further study in larger cohorts of patients, which will open up the possibility of using drugs acting on this link of pathogenesis to treat lung damage in patients with COVID-19.

## 1. Introduction

A new coronavirus infection (NCI), COVID-19 (coronavirus disease 2019), induced by the RNA virus severe acute respiratory syndrome coronavirus-2 (SARS-CoV-2) of the *Coronaviridae* family emerged at the end of 2019 and has caused about 700 million cases of the disease [1]. A critical form of COVID-19 is a cytokine storm, leading to the development of acute respiratory distress syndrome (ARDS), multiple organ failure, and even death [2,3]. The entry of coronavirus into the human body activates innate immune cells, including mast cells (MCs), already known for their role in the development of allergic reactions, infectious and inflammatory processes, the pathogenesis of bronchial asthma, chronic obstructive pulmonary disease (COPD), and other diseases. The activation and degranulation of MCs cause both the release of mediators deposited in them and de novo synthesis, including numerous cytokines and chemokines [4]. Excessive amounts of cytokines—tumor necrosis factor alpha (TNF-α), interferon-γ, interleukin (IL)-1β, and IL-6—underlie the development of a cytokine storm [5]. The hyper-activation of the immune response in COVID-19 is accompanied by damage to the pulmonary parenchyma and is associated with the development of ARDS. We have recently shown that MCs play a role in both early and late stages of alveolar damage [6]. In addition, COVID-19-specific viral and cytokine-storm-induced endothelial damage occurs, which is an important factor in the pathogenesis of hypoxia and ARDS. The mechanisms described above lead to the development of respiratory failure and the need for respiratory support.

A major role in the effects of MCs is played by their proteases: chymase, tryptase, and carboxypeptidase A3 (CPA3) [7]. Briefly, these proteases are synthesized and stored in the cytoplasmic granules of MCs. MCs’ activation is a complex process initiated by the binding of the IgE antibodies to the MCs’ receptor FcεRI [8]. This triggers the cascade of intracellular signaling events with the ultimate occurrence of MCs’ degranulation and release of different active mediators, including the proteases. There are data showing that tryptase expressed by MCs plays a role in the infection by SARS-CoV-2 [9]. There is knowledge about the role of MCs’ chymase in the production of angiotensin II and its contribution to the recruitment of leukocytes, and, consequently, the maintenance of the inflammatory process in the endothelium, which may contribute to the development of SARS-CoV-2-associated endothelial dysfunction [10]. There is also growing evidence of CPA3 involvement in the pathogenesis of COVID-19 [11].

The above mechanisms of the potential participation of MCs and their proteases in the pathogenesis of COVID-19 indicate their important role in the development of lung damage. The purpose of this study was to determine the significance of MCs and their proteases chymase, tryptase, and CPA3 in the pathogenesis of lung damage in patients with COVID-19.

## 2. Materials and Methods

### 2.1. Patients and Study Design

We analyzed 407 patients with COVID-19 and positive results of a polymerase chain reaction (RT-PCR) on the nasal/oropharyngeal swab hospitalized between 2 September 2021 and 3 June 2022 at Voronezh Municipal Clinical Emergency Hospital No. 1 and Voronezh Regional Clinical Hospital No. 1. Medical records of patients were carefully reviewed for exclusion criteria. Exclusion criteria were the presence of other (except COVID-19) confirmed infectious lung diseases (pneumonia of other etiologies, tuberculosis, etc.), the presence of chronic respiratory diseases (bronchial asthma, COPD, occupational lung diseases), the presence of a pulmonary embolism and oncological including oncohematological diseases, the presence of hydrothorax, chronic kidney disease (CKD) above C2 in terms of the glomerular filtration rate (GFR), hepatitis, liver cirrhosis, diabetes mellitus, and history of smoking. Inclusion criteria were a diagnosis of COVID-19 with a positive RT-PCR result on the nasal/oropharyngeal swab, bilateral pneumonia, ARDS, and an age of 18–75 years. A total of 212 patients were excluded because of exclusion criteria or not meeting inclusion criteria. A total of 55 patients, who died as a result of critical COVID-19, were included in the main group (Figure 1).

Autopsy specimens were collected from 55 patients with COVID-19. This study was approved by the Ethics Committee of Voronezh State Medical University (Protocol #8, 17 November 2021) and was conducted with written consent from the patient’s family members, following the regulations issued by the Helsinki Declaration. Autopsy specimens were collected within an autopsy containment unit, which meets the biosafety requirements. A group (*n* = 30) of consecutive consented autopsy cases during the same period, negative for SARS-CoV-2 infection, was used as a control group. 

### 2.2. Laboratory Analyses

This study analyzed the results of laboratory research methods: the first taken upon admission, and the last during a general blood test (with determination of the leukocyte formula, platelets, erythrocytes, hemoglobin, erythrocyte sedimentation rate (ESR)) and blood coagulogram with the determination of the activated partial thromboplastin time (aPTT), prothrombin index (PTI), fibrinogen, international normalized ratio (INR); in addition, patients were assessed for C-reactive protein (CRP).

### 2.3. Histological Analyses

After the death of the patients of the main group, a representative section of the pulmonary parenchyma was collected within 24 h at the pathology departments of Voronezh Municipal Clinical Emergency Hospital No. 1 and Voronezh Regional Clinical Hospital No. 1. A similar step took place in the control group on the basis of the Voronezh Regional Bureau of Forensic Medicine. Autopsy lung material was fixed in 10% neutral buffered formalin and embedded in paraffin with sections 5 µm thick for staining with hematoxylin and eosin and Giemsa staining and 2 µm thick for an immunohistochemical analysis. Immunohistochemical staining was performed according to a standard protocol to detect tryptase, chymase, and CPA3; identification was carried out using primary mouse antibodies Anti-Mast Cell Tryptase antibody (clone AA1, #ab2378, dilution 1:4000) and Anti-Mast Cell Chymase antibody (#ab233103, dilution 1:1000) and rabbit antibodies to CPA3 from Abcam (#ab251685, dilution 1:1000).

The analysis of micropreparations was carried out on the basis of the Research Institute of Experimental Biology and Medicine of the Voronezh State Medical University named after N.N. Burdenko using an Axio Imager.A2 microscope (Carl Zeiss, Oberkochen, Germany) and image processing using the ZEN 2.3 program (Carl Zeiss, Oberkochen, Germany). MCs were calculated using a ×40 lens with an analysis of at least 50 fields of view. The data are presented per 1 mm^2^ calculated using the following formula: 1,000,000× the total number of cells in 50 fields of view/the total of the area fields of view. An analysis of microslides included counting of chymase-, tryptase-, and CPA3-positive MCs with distribution according to the presence of degranulation.

### 2.4. Statistical Analysis

The results were subjected to statistical processing using the Statgraphics Centurion XV.I, version 15.1.2 (Statgraphics Technologies, Inc., The Plains, VA, USA). The normality of data distribution was assessed using normalized coefficients of kurtosis and skewness, as well as the Shapiro–Wilk test. Quantitative data, taking into account non-normal distribution, are presented in the form of Me [V0.25; V0.75], where Me is the median (Me1 is the median in the main group, Me2 is the median in the control group); V0.25 and V0.75 are the lower and upper quartiles. The significance of differences between the main and control groups was assessed using the Mann–Whitney test. A correlation analysis was performed using Spearman’s correlation coefficient. Differences were considered significant at *p* < 0.05.

## 3. Results

Our study included 55 patients with COVID-19: 29 men (53%) and 26 women (47%) with a median age of 67 [62; 71] years. The median duration of the disease was 15 [12; 22.5] days, and the median duration of hospitalization was 9 [5; 14.5] days. The patients had a history of comorbid conditions including arterial hypertension, coronary heart disease (CHD), ischemic brain stroke, chronic heart failure (CHF), obesity, CKD up to stage 2 according to the level of the GFR (Table 1). Among the patients of the main group, non-invasive ventilation (NIV) was performed in 14 patients (25%), invasive mechanical ventilation (IMV) in 30 patients (55%), and only oxygen therapy in 11 patients (20%). The control group was represented by 30 individuals (16 males (53%) and 14 females (47%)), with a median age of 64.5 [58; 70] years, who died from external causes. The study and control groups did not differ significantly in gender, age, and comorbidities (Table 1).

Microscopic changes in lung tissues of patients who died from COVID-19 are represented by characteristic histopathological signs of progressive diffuse damage to the alveoli with excessive thrombosis and a late onset of remodeling of lung tissue and blood vessels. Acute damage to the alveolar–capillary barrier was characterized by damage to alveolar epithelial cells, endothelial cells, and basal cells of the respiratory epithelium and impaired tissue repair processes (Figure 2). Autopsy material from lungs of patients with COVID-19 revealed a wide range of MCs with different protease profiles and degranulation activity (Figure 3).

A comparison of tryptase-, chymase-, and CPA3-positive MCs in the main (COVID-19) and control groups is shown in Table 2, Table 3 and Table 4. Briefly, the content of tryptase-positive MCs in the lungs of patients with COVID-19 was increased compared to the controls, but their degranulation activity was lower. The content of chymase-positive MCs was significantly reduced in patients with COVID-19 in comparison with the controls, while the content of CPA3-positive MCs and their degranulation activity were higher in patients with COVID-19.

### 3.1. The Relationship between MCs and General Blood Test Parameters

We found statistically significant correlations between MCs and general blood test (GBT) indicators. Figure 3 shows the dependencies established for band neutrophils in the GBT taken upon admission to the hospital. Positive correlations were found between the content of tryptase-positive MCs and the level of band neutrophils (Figure 4). No statistically significant relationships were established for the total leukocyte content.

The total number of single CPA3-positive MCs negatively correlates with the content of blood monocytes according to the results of the GBT (r = −0.3840; *p* = 0.044), performed shortly before death. In addition, correlations were found between the content of MCs and the level of eosinophils and basophils (Figure 5).

Positive correlations were identified between the number of CPA3-positive MCs and the level of hemoglobin in the blood (Figure 6).

Interestingly, numerous correlations between MCs’ parameters and the erythrocyte sedimentation rate (ESR) were discovered (Figure 7 and Figure 8). Briefly, the ESR level negatively correlates with the content of tryptase-positive MCs (Figure 7) and chymase-positive MCs (Figure 8).

On the contrary, positive correlations with the level of ESR were established for the relative content of the total number of co-adjacent tryptase-positive MCs, as well as co-adjacent tryptase-positive MCs without signs of degranulation (*p* = 0.010, r = 0.345 and *p* = 0.015, r = 0.327, respectively).

### 3.2. MCs and C-Reactive Protein

We investigated the correlations between MCs’ indicators and the CRP level. No statistically significant relationships were found between MCs’ indicators and the CRP level.

### 3.3. MCs and Coagulogram Parameters

Correlations between coagulogram parameters and MCs are presented in Table 5, Table 6 and Table 7. In general, positive correlations were revealed between the aPTT level and the content of non-degranulating co-adjacent tryptase-positive and total co-adjacent tryptase-positive MCs, single degranulating chymase-positive MCs, degranulating CPA3-positive MCs, and the total number of CPA3-positive MCs. In addition, a negative correlation was found between the total number of CPA3-positive MCs and the blood INR level and a positive correlation between INR and total, single, and co-adjacent non-degranulating chymase-positive MCs. Finally, there was a negative correlation between the level of fibrinogen and single tryptase-positive MCs, while there was a positive correlation between this parameter and total co-adjacent and co-adjacent non-degranulating tryptase-positive MCs. Further, the level of fibrinogen negatively correlates with the total and single content of chymase-positive MCs.

## 4. Discussion

The absolute and relative total number of tryptase-positive MCs, as well as single tryptase-positive MCs, in patients of the COVID-19 group turned out to be greater than in the control group; however, degranulation activity is lower, which confirms the lower content in the COVID-19 group of single and co-adjacent tryptase-positive MCs with signs of degranulation, as well as fragments of tryptase-positive MCs as a product of degranulation. Reduced degranulation activity of tryptase-positive MCs was already detected in our previous study, but we did not find increased content of tryptase-positive MCs and decreased content of chymase-positive MCs in the previous study [12]. This may be due to the exclusion of patients with diabetes in the present study.

Other authors also revealed an increased representation of MCs in the perivascular and septal spaces in autopsy material from the lungs of patients with COVID-19 [13]. Increased numbers of activated MCs and neutrophils in bronchoalveolar lavage fluid were also found in patients with COVID-19 [14].

Gebremeskel S. et al. identified elevated levels of chymase, β-tryptase, and CPA3 in the blood serum of patients with COVID-19, indicating the systemic activation of MCs [15]. Similar results were obtained by Tan J. and co-authors, who determined an increased level of serum chymase in patients with COVID-19 [16]. However, in our study, the total number of chymase-positive MCs and the number of single chymase-positive MCs both with and without signs of degranulation, as well as their fragments, were statistically significantly lower in the main (COVID-19) group compared to the control. At the same time, many studies have established the pro-inflammatory effects of MCs’ chymase due to the activation of cytokines and growth factors, IL-1β, IL-8, IL-18, transforming growth factor-β, endothelin-1 and -2, neutrophil-activating protein-2, and others, which leads to the recruitment of granulocytes, lymphocytes, and monocytes into the tissue microenvironment [17], as well as the role of chymase in the processes of fibrillogenesis, collagen formation, and, ultimately, fibrosis [18].

Furthermore, in the lung tissues of our patients with COVID-19, a statistically significantly greater representation of CPA3-positive MCs was found, including CPA3-positive MCs with signs of degranulation and adjacent CPA3-positive MCs, compared to the control group (*p* = 0.01; *p* < 0.001; *p* < 0.001, respectively). An increased level of CPA3 in the blood serum of patients with COVID-19, regardless of Gebremeskel S. et al., was found by Soria-Castro R. et al. [19].

We found positive correlations between the total content of tryptase-positive MCs, as well as single tryptase-positive MCs, and single tryptase-positive MCs with signs of degranulation and the level of band neutrophils. It was previously known that MCs participate in the active recruitment of neutrophils to the site of inflammation, which largely determines their pro-inflammatory effect. Moreover, there are studies indicating a correlation between the blood level of CPA3-MCs in patients with COVID-19 and the level of neutrophils in the blood, which also confirms the presence of not only a local effect of MCs on the chemotaxis of neutrophils to the site of inflammation, but also a systemic effect on the level of neutrophils in peripheral blood [19]. We also found that the relative content of co-adjacent tryptase-positive MCs with signs of degranulation, as well as the absolute content of co-adjacent CPA3-positive MCs, positively correlates with the content of eosinophils in the last GBT. Previously, other authors reported the effect of MCs’ tryptase on the activation status of eosinophils by inducing the release of eosinophil peroxidase and beta-hexosaminidase from peripheral blood eosinophils [20]. In addition, the involvement of MCs in the pathogenesis of other diseases involving eosinophils was previously reported: eosinophilic esophagitis, bronchial asthma, chronic rhinosinusitis, etc. [21,22,23].

A positive correlation of the relative content of co-adjacent tryptase-positive MCs with signs of degranulation with the content of basophils in the GBT could hypothetically be due to activating signals common to MCs and basophils: through Fc epsilon RI receptors; complement fragments C3a, C4a, and C5a; mediators from activated neutrophils; or some neurotransmitters.

Negative correlations were found between the absolute content of single CPA3-positive MCs in the autopsy material of the lungs of patients with COVID-19 and the content of blood monocytes. It is known that SARS-CoV-2-infected monocytes, macrophages, and MCs produce pro-inflammatory cytokines and chemokines that promote the development of local tissue inflammation and a systemic response in the form of a cytokine storm. The low expression of angiotensin-converting enzyme 2 (by monocytes/macrophages) in patients with COVID-19 may also contribute to the development of pathological reactions due to the pro-inflammatory properties of angiotensin II and dysfunction of the renin–angiotensin system. Both local tissue inflammation and a cytokine storm play a fundamental role in the development of complications associated with COVID-19, such as ARDS, which is the leading cause of death in these patients [24].

Positive correlations were revealed between the total number of CPA3-positive MCs and the number of CPA3-positive MCs with signs of degranulation and blood hemoglobin levels. The effect of SARS-CoV-2 on the structural membrane homeostasis of erythrocytes at the protein and lipid levels is known. Red blood cells of patients with COVID-19 show increased levels of glycolytic intermediates, which is accompanied by the oxidation and fragmentation of membrane proteins. Red blood cells may be unable to respond to changes in hemoglobin oxygenation as they move from the lungs into the bloodstream and may have a reduced ability to transport and deliver oxygen [25]. Laboratory tests in patients with COVID-19 show a decrease in hemoglobin concentration and a pathologically increased ferritin concentration [26]. Previously, we obtained data regarding the relationship between MC tryptase and hemoglobin and erythrocyte levels. Strong negative correlations were found between the total number of tryptase-positive MCs, as well as the number of tryptase-positive MCs with signs of degranulation, and the content of red blood cells, which may form a hypothesis about the potential connection of MCs and the products of their degranulation with the development of anemia. At the same time, the negative correlation of the number of tryptase-positive MCs without signs of degranulation with the hemoglobin content introduces some contradiction [12].

A positive correlation of the aPTT level in the last analysis with the relative content of co-adjacent tryptase-positive MCs without signs of degranulation may indicate the depletion of degranulation activity. At the same time, positive correlations were found between the aPTT level and the relative content of single degranulating chymase-positive MCs, the absolute content of CPA3-positive MCs with signs of degranulation, and the total number of CPA3-positive MCs. In addition, a negative correlation was found between the total number of CPA3-positive MCs and the blood INR level. However, it is unclear whether the degranulation activity of MCs has an effect on the systemic coagulation status or whether this effect is associated with the use of anticoagulant therapy. MCs in the perivascular space of small pulmonary vessels are known to release heparin to prevent thrombosis. However, it is believed that heparin of MCs does not have a significant systemic effect. On the other hand, there is information about the role of heparin of MCs, especially in the composition of complexes with tryptase and chymase, in the regulation of blood coagulation [27]. The blood fibrinogen level determined upon admission negatively correlates with the relative content of single tryptase-positive MCs, as well as with the absolute total content of chymase-positive MCs. This is quite consistent with the known phenomenon of coagulopathy, specifically hypofibrinogenemia, in patients with hyperinflammation in COVID-19.

It is worth noting that in this study, we did not perform experiments regarding the D-dimer. Numerous studies showed that D-dimer is a marker of coagulation and thrombosis in COVID-19 and high D-dimer is associated with mortality in COVID-19 [28,29]. Taking this into account, we consider the possibility of further studies including D-dimer and recommend it to other authors.

Additionally, in future studies, it is also planned to conduct a Western blot to detect MC proteases, which, in addition to an immunohistochemical analysis, will ensure maximum accuracy and objectivity of the data.

## 5. Conclusions

Summarizing the above results, it is worth noting that the content of tryptase-positive MCs in the lungs of patients with COVID-19 is significantly higher than in control individuals; however, degranulation activity is lower, which may be either an initial phenomenon or a consequence of the exhaustion of degranulation processes while the disease progresses. It is not possible to clarify this, because the histological sample reflects the picture only at a specific point in time. Carrying out intravital biopsies over time could clarify the picture and it is a possible promising direction for research. Indicators of chymase-positive MCs in the lungs of patients with COVID-19 are presented in statistically significantly lower numbers than in controls, while the content of CPA3-positive MCs and their degranulation activity are higher in patients with COVID-19.

Relationships between the content of various subpopulations of MCs and the degree of their degranulation activity in the lungs of patients with COVID-19 and the levels of band neutrophils, basophils, eosinophils, and monocytes in the peripheral blood; hemoglobin levels; and ESR have been established. The result of no established correlations between MCs and the CRP level is contradictory. A prospective direction for further research may be the numerous connections between aPTT, fibrinogen, and INR and various indicators of MCs, indicating the influence of MCs not only on local, but also, possibly, systemic coagulation status. The question remains open whether the degranulation activity of MCs has an effect on the systemic coagulation status or whether this effect is associated solely with the use of anticoagulant therapy.

Thus, the identified patterns shed light on the numerous and diverse mechanisms of the participation of MCs and their proteases in the pathogenesis of COVID-19. At the same time, the issue requires further study in larger cohorts of patients, which will open up the possibility of using drugs acting on this link of pathogenesis to treat lung damage in patients with COVID-19, reduce hospitalization and disability, and improve disease outcomes. Finally, future comprehensive in vitro and in vivo studies are warranted in order to reveal the profound molecular mechanisms and to identify the potential therapeutic options.

## Figures and Tables

**Figure 1 cells-13-00711-f001:**
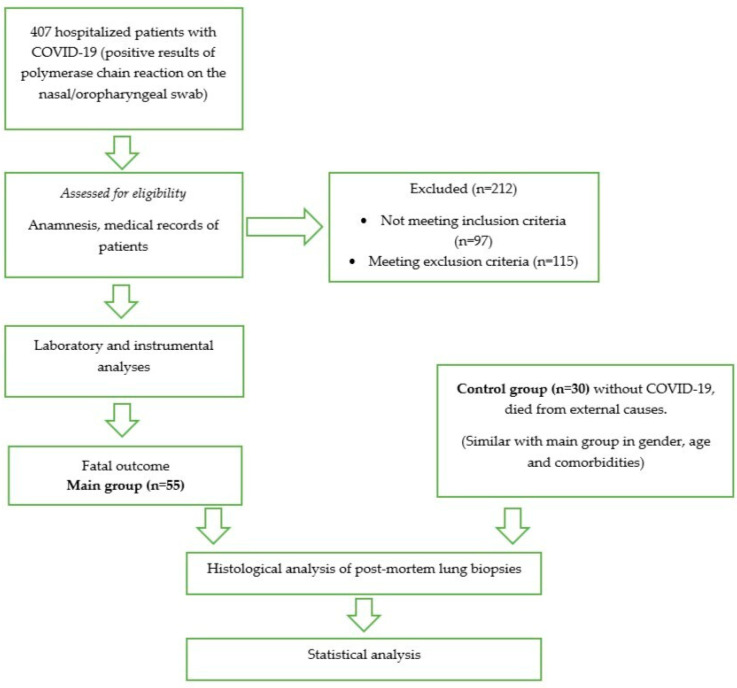
Flowchart of study design.

**Figure 2 cells-13-00711-f002:**
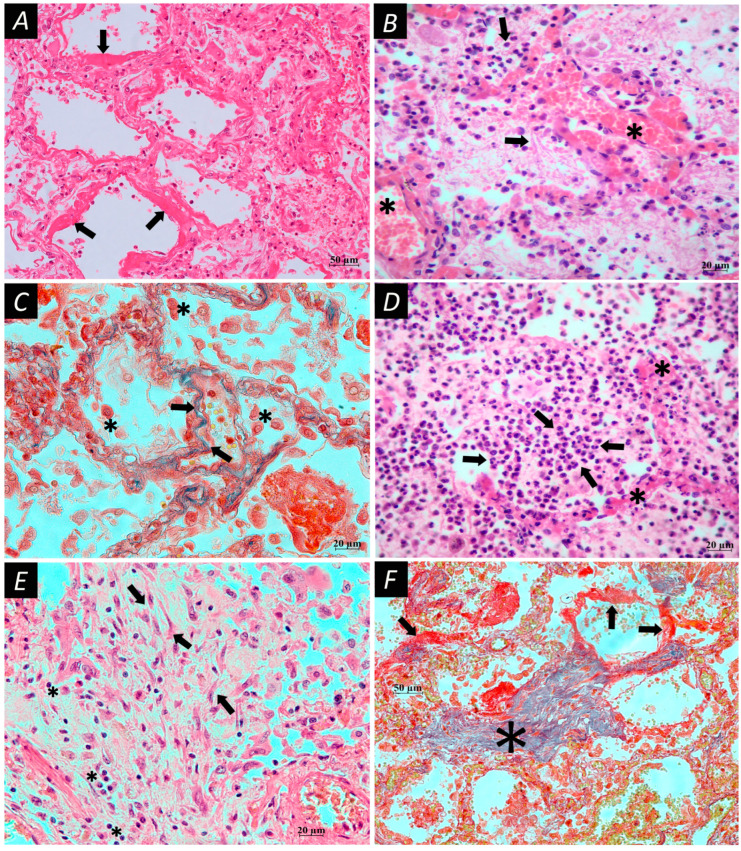
Microscopic changes in lung tissue in patients who died from COVID-19. (**A**) The walls of the alveoli with deposits of fibrin masses are hyaline membranes (arrows); in the lumen of the alveoli, dissociated cellular elements are the desquamated epithelium. (**B**) Acute non-COVID-19 pneumonia for comparison: sharply full-blooded vessels of the alveoli (star), and masses of fibrin and inflammatory cells in the lumen (arrows). (**C**) Desquamated respiratory epithelial cells in the lumen of the alveoli (stars); thin areas of fibrosis in the artery wall and partially in the septa of the alveoli (blue color, arrows). (**D**) Acute purulent pneumonia (for comparison), a lot of leukocytes in the lumen of the alveoli (arrows), full-blooded vessels of the alveolar septa (stars). (**E**) The fibrosis site, bundles, and separate fibroblasts (arrows); weakly expressed lymphoplasmocytic inflammatory infiltration (stars). (**F**) A combination of diffuse alveolar damage in the early stage (hyaline membranes, arrows) and areas of fibrosis; special coloring reveals connective tissue fibers (blue color, star). Technique: (**A**,**B**,**D**,**E**)—hematoxylin and eosin; (**C**,**F**)—picro Mallory staining. Scale bar: (**A**,**F**)—50 µm, (**B**–**E**)—20 µm.

**Figure 3 cells-13-00711-f003:**
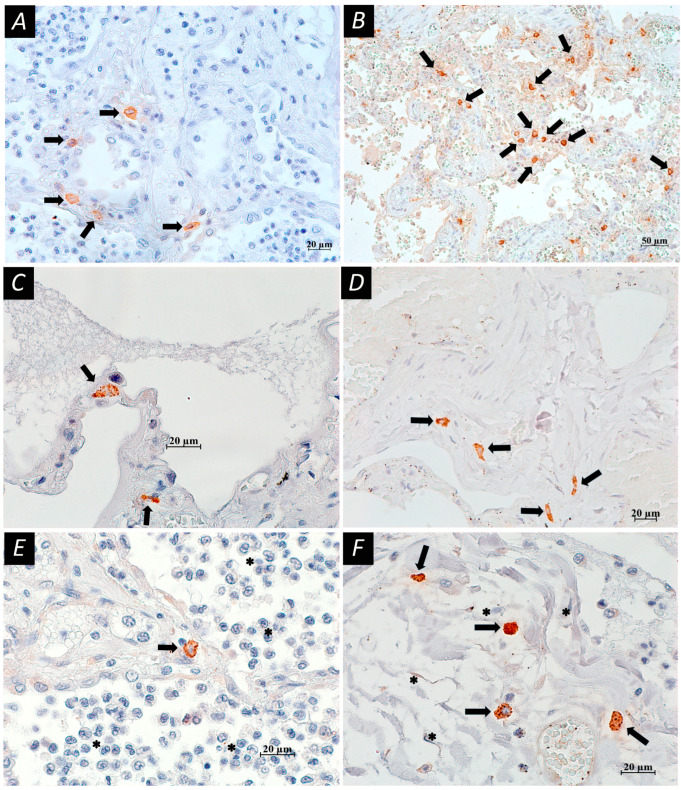
Histotopography and cellular interactions of mast cell (MC) proteases in lung tissues of patients who died from COVID-19. Immunohistochemical reaction with antibodies to tryptase (**A**,**B**), carboxypeptidase (**C**,**E**), chymase (**D**,**F**); nuclei were counterstained with Mayer’s hematoxylin. (**A**) Massive infiltration of lung structures by tryptase-positive MCs (arrows). (**B**) Perivascular location of tryptase-positive MCs with signs of degranulation (arrows). (**C**) Two mast cells (arrows) in the alveolar septum with degranulation phenomena. (**D**) Group of mast cells (arrows) with signs of degranulation in fibrotic area. (**E**) Mast cell in the alveolar septum (arrow), and numerous desquamated cells (stars) in the lumen of the alveoli. (**F**) Accumulation of mast cells (arrows) and fibroblasts (stars) in lung tissue; granules of mast cells are clearly visible without signs of degranulation. Scale bar: (**B**)—50 µm, (**A**,**C**–**F**)—20 µm.

**Figure 4 cells-13-00711-f004:**
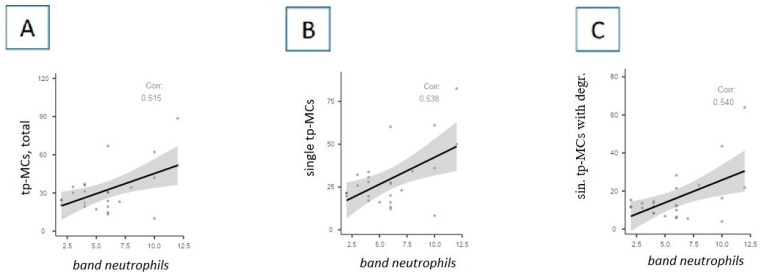
Correlations between the content of band neutrophils in peripheral blood and indicators of tryptase-positive MCs in autopsy material from the lungs of patients with COVID-19. (**A**) Absolute (per mm^2^) total content of tryptase-positive MCs positively correlates with the relative content of band neutrophils in the general blood test (GBT) (*p* = 0.008; r = 0.515). (**B**) Absolute (per mm^2^) content of single tryptase-positive MCs positively correlates with the relative content of band neutrophils in the GBT (*p* = 0.005; r = 0.538). (**C**) Absolute (per mm^2^) content of single tryptase-positive MCs with signs of degranulation positively correlates with the relative content of band neutrophils in the GBT (*p* = 0.005; r = 0.540). Legend: tp-MCs, total—absolute (per mm^2^) total content of tryptase-positive mast cells; single tp-MCs—absolute (per mm^2^) content of single tryptase-positive mast cells; sin. tp-MCs with degr.—absolute (per mm^2^) content of single tryptase-positive mast cells with signs of degranulation.

**Figure 5 cells-13-00711-f005:**
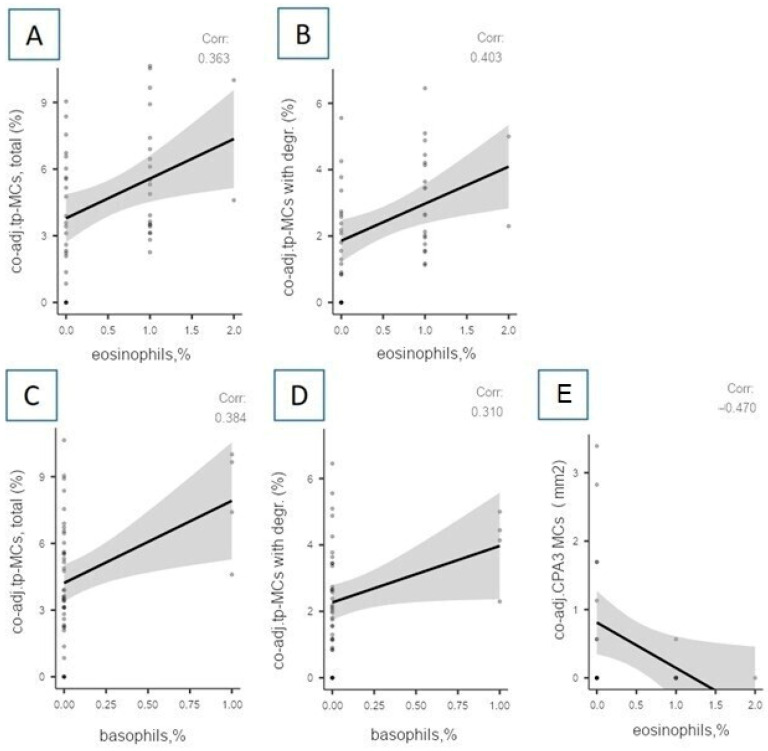
Correlations between the content of eosinophils and basophils in peripheral blood and indicators of mast cells in autopsy material from the lungs of patients with COVID-19. (**A**) Relative content (%) of co-adjacent tryptase-positive MCs positively correlates with the relative content of eosinophils in the last GBT performed shortly before death (r = 0.363; *p* = 0.013). (**B**) Relative content (%) of co-adjacent tryptase-positive MCs with signs of degranulation positively correlates with the relative content of eosinophils in the last GBT performed shortly before death (r = 0.403; *p* = 0.007). (**C**) Relative content (%) of co-adjacent tryptase-positive MCs positively correlates with the relative content of basophils in the last GBT performed shortly before death (r = 0.384; *p* = 0.01). (**D**) Relative content (%) of co-adjacent tryptase-positive MCs with signs of degranulation positively correlates with the relative content of basophils in the last GBT performed shortly before death (r = 0.310; *p* = 0.048). (**E**) Absolute (mm^2^) content of co-adjacent CPA3-positive MCs correlates negatively with the relative content of eosinophils according to the results of a GBT performed upon admission to the hospital (*p* = 0.015, r = −0.470). Legend: co-adj.tp-MCs with degr. (%)—co-adjacent tryptase-positive mast cells with signs of degranulation; co-adj.tp-MCs, total (%)—co-adjacent tryptase-positive mast cells, total count; co-adj. CPA3 MCs–co-adjacent carboxypeptidase A3-positive mast cells.

**Figure 6 cells-13-00711-f006:**
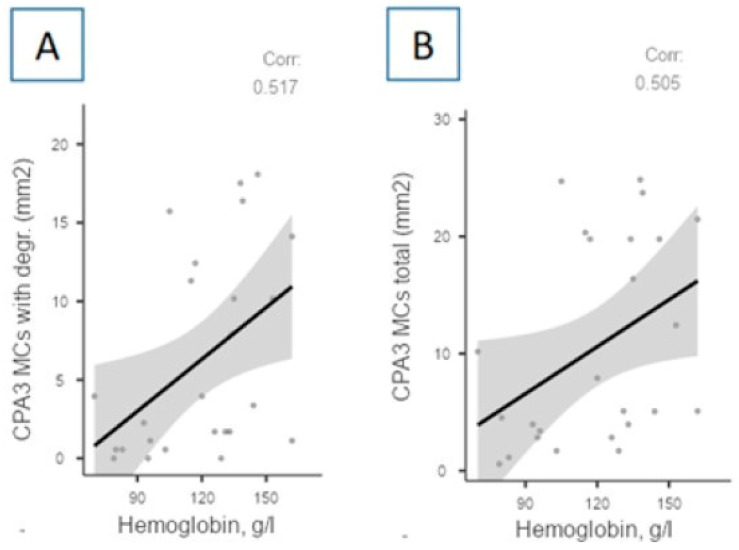
Correlations between the hemoglobin level and indicators of carboxypeptidase A3-positive mast cells in autopsy material from the lungs of patients with COVID-19. The level of hemoglobin in the blood according to the results of the last GBT performed on the patient shortly before death correlates positively with the absolute (mm^2^) level of CPA3-positive MCs with signs of degranulation (**A**) and the total absolute (mm^2^) number of CPA3-positive MCs (**B**) (*p* = 0.008, r = 0.517 and *p* = 0.010, r = 0.505, respectively). Legend: CPA3 MCs with degr.—carboxypeptidase A3-positive mast cells with signs of degranulation; CPA3 MCs, total—carboxypeptidase A3-positive mast cells, total count.

**Figure 7 cells-13-00711-f007:**
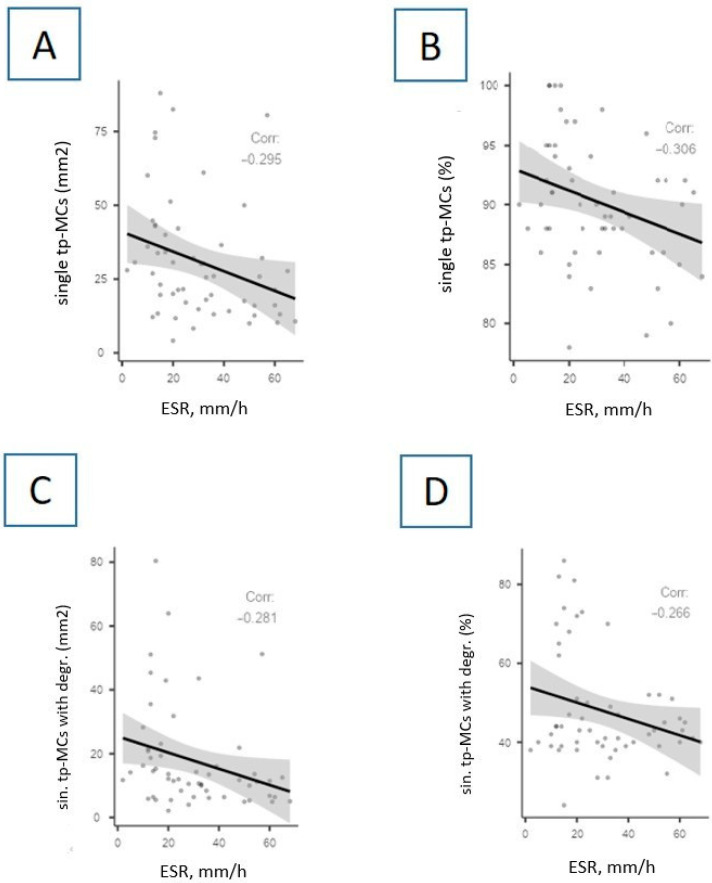
Correlations between the ESR level and the indicators of tryptase-positive MCs in autopsy material of the lungs of patients with COVID-19. The ESR level negatively correlates with the absolute (**A**) and relative (**B**) content of single tryptase-positive MCs (*p* = 0.029, r = −0.295 and *p* = 0.023, r = −0.306, respectively), as well as with the number of single tryptase-positive MCs with signs of degranulation in absolute (**C**) and relative (**D**) equivalents (*p* = 0.038, r = −0.281 and *p* = 0.005, r = −0.266, respectively). Legend: ESR—erythrocyte sedimentation rate; single tp-MCs—single tryptase-positive mast cells; sin.tp-MCs with degr.—single tryptase-positive mast cells with signs of degranulation.

**Figure 8 cells-13-00711-f008:**
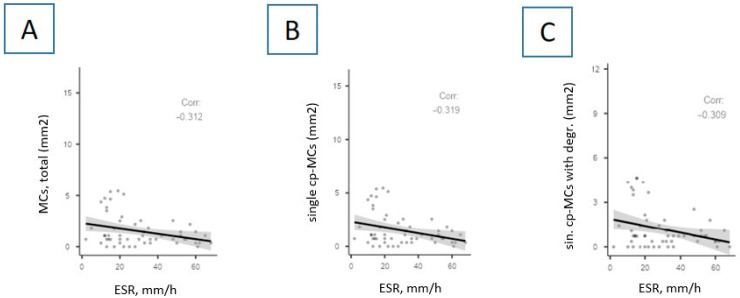
Correlations between the ESR level and the indicators of chymase-positive MCs in autopsy material of the lungs of patients with COVID-19. The ESR level negatively correlates with the total number of chymase-positive MCs (**A**) (per mm^2^) (*p* = 0.02, r = −0.312), the absolute number of single chymase-positive MCs (**B**) (*p* = 0.018, r = −0.319), and the absolute number of single chymase-positive MCs with signs of degranulation (**C**) (*p* = 0.022, r = −0.309). Legend: ESR—erythrocyte sedimentation rate; cp-MCs, total—total content of chymase-positive mast cells; single cp-MCs—single chymase-positive mast cells; sin. cp-MCs with degr.—single chymase-positive mast cells with signs of degranulation.

**Table 1 cells-13-00711-t001:** Comparative characteristics of the main (COVID-19) and control groups.

Parameters	COVID-19 Group	The Control Group
Number of subjects	55	30
Male, n (%)	29 (53)	16 (53)
Female, n (%)	26 (47)	14 (47)
Age, years	67 [62; 71]	64.5 [58; 70]
PCR SARS-CoV-2 «+», n (%)	55 (100)	0 (0)
Bilateral pneumonia, n (%)	55 (100)	0 (0)
ARDS, n (%)	55 (100)	0 (0)
Arterial hypertension, n (%)	45 (82)	25 (83)
Coronary heart disease, n (%)	6 (11)	4 (13)
Ischemic brain stroke, n (%)	10 (18)	5 (17)
Chronic heart failure (CHF), n (%)	15 (27)	9 (30)
Obesity, n (%)	14 (25)	8 (27)
I degree	11 (20)	5 (17)
II degree	1 (2)	1 (3)
III degree	2 (4)	2 (7)
Chronic kidney disease, n (%) (C1–C2 stages)	11 (20)	7 (23)

Data are presented as absolute values (%) or median [lower and upper quartiles]. ARDS: acute respiratory distress syndrome.

**Table 2 cells-13-00711-t002:** Differences in the content of tryptase-positive mast cells (MCs) in the main (COVID-19) and control groups.

	COVID-19 Group, Number of MCs per mm^2^	The Control Group, Number of MCs per mm^2^	*p*-Test Value	COVID-19 Group, % MCs	The Control Group, % MCs	*p*-Test Value
Single tryptase + MCs without degranulation	10.4 [7.13; 18.9]	1.78 [1.17; 2.22]	<0.001	46.43 [35.31; 50.32]	7.97 [6.44; 11.1]	<0.001
Single tryptase + MCs with degranulation	12.0 [6.61; 20.12]	14.78 [10.41; 19.91]	0.408	43.02 [39.51; 51.11]	72.13 [65.21; 77.21]	<0.001
Single tryptase + MCs total	26.0 [16.1; 38.3]	16.78 [12.82; 22.5]	0.005	89.84 [86.91; 93.92]	81.94 [73.1; 86.72]	<0.001
Co-adjacent tryptase + MCs without degranulation	0.53 [0.07; 0.8]	0 [0; 0.44]	<0.001	2.02 [0.23; 3.81]	0 [0; 0.61]	<0.001
Co-adjacent tryptase + MCs with degranulation	0.53 [0.27; 0.8]	0.56 [0.44; 1.33]	0.061	2.13 [1.02; 3.45]	3.77 [2.74; 5.23]	<0.001
Co-adjacent tryptase + MCs total	1.07 [0.53; 1.71]	0.89 [0.44; 1.50]	0.391	4.04 [2.71; 6.64]	4.6 [3.71; 5.48]	0.858
Fragments of tryptase + MCs	1.33 [0.67; 2.13]	2.56 [1.39; 5.72]	<0.001	5.36 [3.17; 6.87]	11.7 [9.84; 22.02]	<0.001
Total amount of tryptase + MCs	30.1 [18.2; 41.6]	22.1 [15.2; 27.9]	0.033	-	-	-

Data are presented as Me [V0.25; V0.75], where Me is the median, and V0.25 and V0.75 are the lower and upper quartiles.

**Table 3 cells-13-00711-t003:** Differences in the content of chymase-positive mast cells (MCs) in the main (COVID-19) and control groups.

	COVID-19 Group, Number of MCs per mm^2^	The Control Group, Number of MCs per mm^2^	*p*-Test Value	COVID-19 Group, % MCs	The Control Group, % MCs	*p*-Test Value
Single chymase + MCs without degranulation	0 [0; 0.36]	0.44 [0.13; 0.89]	<0.001	0 [0; 33.33]	23.9 [16.81; 33.33]	0.030
Single chymase + MCs with degranulation	0.73 [0.36; 1.45]	1.2 [0.47; 2.76]	0.048	75 [25; 100]	65.1 [55.33; 71.51]	0.123
Single chymase + MCs total	1.07 [0.36; 1.82]	1.8 [0.7; 3.43]	0.034	100 [100; 100]	92.31 [81.81; 100]	<0.001
Co-adjacent chymase + MCs without degranulation	0 [0; 0]	0 [0; 0]	0.184	0 [0; 0]	0 [0; 0]	0.184
Co-adjacent chymase + MCs with degranulation	0 [0; 0]	0 [0; 0.08]	0.011	0 [0; 0]	0 [0; 1.5]	0.013
Co-adjacent chymase + MCs total	0 [0; 0]	0 [0; 0.08]	0.011	0 [0; 0]	0 [0; 1.5]	0.013
Fragments of chymase + MCs	0 [0; 0]	0.13 [0; 0.67]	<0.001	0 [0; 0]	7.69 [0; 12.9]	<0.001
Total amount of chymase + MCs	1.07 [0.54; 2]	1.87 [0.8; 4.57]	0.013	-	-	-

Data are presented as Me [V0.25; V0.75], where Me is the median, and V0.25 and V0.75 are the lower and upper quartiles.

**Table 4 cells-13-00711-t004:** Differences in the content of CPA3-positive mast cells (MCs) in the main (COVID-19) and control groups.

	COVID-19 Group, Number of MCs per mm^2^	The Control Group, Number of MCs per mm^2^	*p*-Test Value	COVID-19 Group, % MCs	The Control Group, % MCs	*p*-Test Value
Single CPA3 + MCs without degranulation	4.35 [1.81; 7.21]	3.48 [2.32; 4.11]	0.796	47.22 [36.33; 66.66]	60.38 [52.38; 80.14]	0.072
Single CPA3 + MCs with degranulation	6.49 [1.12; 11.11]	2.07 [1.01; 2.71]	<0.001	52.78 [33.33; 63.66]	39.61 [19.85; 47.62]	0.062
Co-adjacent CPA3 + MCs total	0.41 [0.11; 0.52]	0.06 [0; 0.11]	<0.001	0 [0; 0.03]	0 [0; 0]	0.058
Fragments of CPA3+ MCs	1.52 [0.12; 2.31]	1.27 [0.57; 1.71]	0.779	10 [0; 19.0]	23.05 [7.68; 34.5]	0.099
Total amount of CPA3 + MCs	10.84 [3.51; 19.82]	5.56 [4.22; 6.81]	0.01	-	-	-

Data are presented as Me [V0.25; V0.75], where Me is the median, and V0.25 and V0.75 are the lower and upper quartiles.

**Table 5 cells-13-00711-t005:** Results of correlation analysis of coagulogram parameters and tryptase-positive MCs’ parameters in autopsy lung material.

Parameters	Single Tryptase + MCs without Degranulation, %	Single Tryptase + MCs with Degranulation, %	Single Tryptase + MCs Total, %	Co-AdjacentTryptase + MCs without Degranulation, %	Co-Adjacent Tryptase + MCs with Degranulation, %	Co-Adjacent Tryptase + MCs Total, %	Fragments of Tryptase + MCs, %
aPTT No. 1 **, s	0.070	−0.137	−0.191	0.276 *	0.161	0.283 *	0.048
aPTT No. 2 ***, s	0.087	−0.183	−0.302	0.292	0.222	0.350 *	0.157
Fibrinogen No. 1 **, g/L	0.087	−0.209	−0.353 *	0.471 *	−0.022	0.333 *	0.243
Fibrinogen No. 2 ***, g/L	0.141	−0.221	−0.241	0.232	−0.021	0.167	0.218
INR No. 1. **	−0.093	0.018	−0.150	0.283	−0.056	0.204	0.039
INR No. 2 ***	0.212	−0.231	0.035	0.306	−0.181	0.077	−0.191
Prothrombin index No. 1. **, %	−0.219	0.236	0.066	−0.147	−0.077	−0.115	−0.013
Prothrombin index No. 2 ***, %	−0.122	0.047	−0.349	0.255	0.326	0.311	0.328

* *p* < 0.05; ** upon admission to hospital; *** in the latest analysis. Legend: aPTT—activated partial thromboplastin time, INR—international normalized ratio.

**Table 6 cells-13-00711-t006:** Results of correlation analysis of coagulogram parameters and parameters of chymase-positive MCs in autopsy lung material.

Parameters	Single Chymase + MCs without Degranulation, %	Single Chymase + MCs with Degranulation, %	Single Chymase + MCs Total, % Abs.	Co-Adjacent Chymase + MCs without Degranulation, % Abs.	Co-Adjacent Chymase + MCs with Degranulation, %	Co-Adjacent Chymase + MCs Total, %	Fragments of Chymase + MCs, %	Total Amount of Chymase + MCs, Abs.
aPTT No. 1 **, s	−0.052	0.125	0.131	-	0.016	0.016	−0.086	0.129
aPTT No. 2 ***, s	−0.260	0.368 *	−0.082	-	0.055	0.055	0.095	−0.077
Fibrinogen No. 1 **, g/L	0.186	−0.146	−0.404 *	-	0.046	0.046	0.120	−0.398 *
Fibrinogen No. 2 ***, g/L	0.180	−0.148	−0.283	-	−0.092	−0.092	-	−0.288
INR No. 1 **	−0.091	0.037	0.074	-	0.118	0.675	-	0.096
INR No. 2 ***	−0.026	0.081	0.728 *	0.744 *	-	-	-	0.728 *
Prothrombin index No. 1 **, %	0.022	−0.285	−0.159	-	-	-	-	−0.159
Prothrombin index No. 2 ***, %	−0.146	0.146	0.151	-	-	-	-	0.151

* *p* < 0.05; ** upon admission to hospital; *** in the latest analysis. Legend: aPTT—activated partial thromboplastin time, INR—international normalized ratio.

**Table 7 cells-13-00711-t007:** Results of correlation analysis of coagulogram parameters and parameters of CPA3-positive MCs in autopsy lung material.

Parameters	Single CPA3 + MCs without Degranulation, Abs.	Single CPA3 + MCs with Degranulation, Abs.	Fragments of CPA3 + MCs, Abs.	Co-Adjacent CPA3 + MCs Total, Abs.	Total Number of CPA3 + MCs, Abs.
aPTT No. 1 **, s	0.260	0.304 *	0.335	−0.090	0.375 *
aPTT No. 2 ***, s	0.025	−0.182	0.238	−0.260	−0.003
Fibrinogen No. 1 **, g/L	−0.112	−0.106	0.038	−0.300	−0.102
Fibrinogen No. 2 ***, g/L	−0.297	−0.110	−0.072	0.032	−0.226
INR No. 1 **	−0.257	0.170	0.345	0.112	−0.812 *
INR No. 2 ***	−0.232	0.218	0.418	0.245	−0.453
Prothrombin index No. 1 **, %	0.249	0.068	−0.093	−0.134	0.270
Prothrombin index No. 2 ***, %	0.107	−0.107	0.000	−0.112	0.036

* *p* < 0.05; ** upon admission to hospital; *** in the latest analysis. Legend: aPTT—activated partial thromboplastin time, INR—international normalized ratio.

## Data Availability

The data are available from the corresponding author upon a reasonable request.

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
