# Peer review of "Involvement of Mast Cells in the Pathology of COVID-19: Clinical and Laboratory Parallels"

_cells, 2024, doi:10.3390/cells13080711_

Round 1
Reviewer 1 Report
Comments and Suggestions for Authors
In the provided manuscript, the author emphasizes the significance of mast cells (MCs) and their proteases, including chymase, tryptase, and carboxypeptidase A3 (CPA3), in the progression of lung damage in COVID-19 patients. A previous study with matching content (PMCID: PMC7899934) that is not cited in this manuscript reported that MC activation is not a predominant factor in the inflammatory response to SARS-CoV-2 in patients with MC disorders; however, elevated levels of tryptase were detected in the serum. Correlation analysis with R values does not indicate strong associations between the two independent variables. Further investigations into the role of MCs and it's granular content in COVID-19 pathology through well-designed in vitro and in vivo models are warranted. The current manuscript lacks originality and would benefit from providing mechanistic insights to bridge the existing knowledge gap.
Comments on the Quality of English LanguageAcceptable
Author Response
Reviewer 1
Q1. In the provided manuscript, the author emphasizes the significance of mast cells (MCs) and their proteases, including chymase, tryptase, and carboxypeptidase A3 (CPA3), in the progression of lung damage in COVID-19 patients.
R1. We thank to the reviewer for the nice overview of our study.
Q2. A previous study with matching content (PMCID: PMC7899934) that is not cited in this manuscript reported that MC activation is not a predominant factor in the inflammatory response to SARS-CoV-2 in patients with MC disorders; however, elevated levels of tryptase were detected in the serum.
R2. We thank to the reviewer for this important comment. We would like to explain that in the mentioned study, the authors report the impact of SARS-CoV-2 infection in 28 patients with clonal (n = 24) MCs disorders including mastocytosis and in patients with clinical symptoms of MCs activation and elevated baseline serum tryptase with hereditary alpha tryptasemia. In our study, we report some clinical and histological parallels between mast cells and their degranulation and laboratory parameters in the context of COVID-19 without additional MCs disorders. Therefore, these two studies are not directly comparable.
Q3. Correlation analysis with R values does not indicate strong associations between the two independent variables.
R3. We consider correlations above 0.7 to be strong; correlations between 0.3 and 0.69 are moderate, and those below 0.29 are considered weak. You are right, the correlations we discovered do not meet the criteria for strong ones; most of these correlations are moderate, which is a fairly good indicator in medicine. All of these correlations are statistically significant.
Q4. Further investigations into the role of MCs and it's granular content in COVID-19 pathology through well-designed in vitro and in vivo models are warranted. The current manuscript lacks originality and would benefit from providing mechanistic insights to bridge the existing knowledge gap.
R4. We thank to reviewer for raising up this important question. According to the literature, there is also an increase in the degree of mast cells degranulation during the development of COVID-19, modeled on non-human primates, Chinese rhesus macaques and various species of mice, while the degree of mast cells activity correlates with the severity of the infection. Tan J.Y. et al. (2023) observed widespread degranulation of mast cells during acute and unresolved airway inflammation in SARS-CoV-2-infected mice and non-human primates. Using a mouse model of mast cell deficiency, mast cell-dependent interstitial pneumonitis, hemorrhaging, and edema in the lung were observed during SARS-CoV-2 infection [Tan JY, Anderson DE, Rathore AP, O'Neill A, Mantri CK, Saron WA, Lee CQ, Cui CW, Kang AE, Foo R, Kalimuddin S, Low JG, Ho L, Tambyah P, Burke TW, Woods CW, Chan KR, Karhausen J, St John AL. Mast cell activation in lungs during SARS-CoV-2 infection associated with lung pathology and severe COVID-19. Journal of Clinical Investigation. 2023; 133(19):e149834. doi: 10.1172/JCI149834. PMID: 37561585; PMCID: PMC10541193]. Wu M.L. et al (2021) showed that SARS-CoV-2-triggered mast cell degranulation initiated alveolar epithelial inflammation and lung injury. SARS-CoV-2 challenge induced mast cell degranulation in ACE-2 humanized mice and rhesus macaques, and a rapid mast cell degranulation could be recapitulated with Spike-RBD binding to ACE2 in cells; mast cells degranulation altered various signaling pathways in alveolar epithelial cells, particularly, the induction of pro-inflammatory factors and consequential disruption of tight junctions. Importantly, the administration of clinical mast cell stabilizers for blocking degranulation dampened SARS-CoV-2-induced production of pro-inflammatory factors and prevented lung injury [Wu ML, Liu FL, Sun J, Li X, He XY, Zheng HY, Zhou YH, Yan Q, Chen L, Yu GY, Chang J, Jin X, Zhao J, Chen XW, Zheng YT, Wang JH. SARS-CoV-2-triggered mast cell rapid degranulation induces alveolar epithelial inflammation and lung injury. Signal Transduct Target Therapy. 2021 Dec 17;6(1):428. doi: 10.1038/s41392-021-00849-0. PMID: 34921131; PMCID: PMC8677926]. Despite the existence of the above-mentioned literature, we fully agree that in the future the comprehensive in vitro and in vivo studies are crucially needed, which will reveal the underlying pathological mechanisms and the potential therapeutic strategies. Therefore, we have clearly indicated this point in the revised manuscript version, as a future research direction, as follows (Conclusion, page 18, lines 450-452, marked in red): “Finally, future comprehensive in vitro and in vivo studies are warranted in order to reveal the profound molecular mechanisms and to identify the potential therapeutic options.”
Reviewer 2 Report
Comments and Suggestions for Authors
Dear Authors,
The study investigates the role of mast cells (MCs) and their proteases in the pathogenesis of lung damage in patients with COVID-19. COVID-19, caused by SARS-CoV-2, can lead to severe respiratory complications, including acute respiratory distress syndrome (ARDS), due to a cytokine storm. The study aims to assess the significance of MCs and their proteases (chymase, tryptase, and CPA3) in lung damage associated with COVID-19 through histological and laboratory analyses of autopsy specimens.
The study employs a comprehensive approach, combining histological analyses of autopsy specimens with laboratory investigations to assess the involvement of MCs and their proteases in lung damage.
Author Response
Reviewer 2
Q1. Dear Authors,
The study investigates the role of mast cells (MCs) and their proteases in the pathogenesis of lung damage in patients with COVID-19. COVID-19, caused by SARS-CoV-2, can lead to severe respiratory complications, including acute respiratory distress syndrome (ARDS), due to a cytokine storm. The study aims to assess the significance of MCs and their proteases (chymase, tryptase, and CPA3) in lung damage associated with COVID-19 through histological and laboratory analyses of autopsy specimens.
The study employs a comprehensive approach, combining histological analyses of autopsy specimens with laboratory investigations to assess the involvement of MCs and their proteases in lung damage.
R1. We are thankful to the reviewer´s clear summary of our study and we are grateful to the reviewer´s full support and nice comments about our manuscript.
Reviewer 3 Report
Comments and Suggestions for Authors
In this experimental work, Dr. Budnevsky and colleagues explored the role of mast cells and the degranulation products (proteases) in COVID-19. Overall, this is an exciting and nicely done study. Nonetheless, I do have some questions and comments:
1) "We have previously described the potential role of mast cells (MCs) in the pathology of coronavirus disease 2019 (COVID-19)." I don't think this statement is necessary. Please remove it from the abstract.
2) Instead, add some background information supporting the reasoning why this study is important in the abstract (before the objective).
3) Please specify the severity of COVID-19 included in this study before death: mild, moderate, severe, or critical.
4) Why didn't the authors check D-dimer in this study? It has been shown in numerous studies that d-dimer is a marker of coagulation and thrombosis in COVID-19 (PMID: 37092518) and high D-dimer is associated with mortality in COVID-19 (PMID: 33395786).
5) What was expected from procalcitonin measurement in this study? Is it also increased in viral infections such as COVID-19?
6) Line 69: I think the authors need to explain briefly how these proteases are formed and dispatched from MCs. This publication might be helpful (https://doi.org/10.1016/j.mbm.2024.100041).
7) Please specify the ethical approval number provided by Voronezh State Medical University in line 89.
8) Is there any separate ethical approval from the hospital where the autopsy was done? If so, please also include it.
9) The exclusion criteria described in line 94 onward were discovered during autopsy or from medical record when the patients were still alive? Please clarify.
10) I think it is very important to provide the flowchart of study design as at the moment, it is quite confusing to understand the step-by-step experimental procedures. For example, which data were taken while the patients were still alive and which were taken after death.
11) How many patients are excluded? due to? Include it in the flowchart
12) Why did the authors use IHC instead of Western Blot or flowcytometry to detect MC proteases? I am afraid that IHC will provide a subjective (operator-dependent) result. Please comment on this important issue and perform additional experiments to confirm the current IHC findings.
13) It is interesting that CVD was not excluded in this study, while it has been shown that MCs are involved in the pathogenesis of CVD including hypertension (PMID: 31295950; 35048969). Why?
14) in Figure 1, I do think that it is better to also show the histological images of lung tissues from people who died not due to COVID-19 side by side as comparison. Please provide.
15) Also, when explaining Figure 1, please provide arrows or something to show for instance the fibrosis, desquamation, etc. Not all of the readers are pathologists.
16) Similarly, add arrows in Figure 2.
17) Please explain the quantification methods from IHC samples done in Tables 2-4.
18) Please explain why in Figure 8 we saw procalcitonin only on 0 or 10 ng/ml? Isn't it weird as it was like a qualitative measurement (present or absent)? I don't think it is appropriate to make a correlation if the data is nominal. Also, explain why only tryptase was compared to procalcitonin?
Comments on the Quality of English LanguageSome corrections are needed
Author Response
Reviewer 3
Q1. In this experimental work, Dr. Budnevsky and colleagues explored the role of mast cells and the degranulation products (proteases) in COVID-19. Overall, this is an exciting and nicely done study. Nonetheless, I do have some questions and comments:
1) "We have previously described the potential role of mast cells (MCs) in the pathology of coronavirus disease 2019 (COVID-19)." I don't think this statement is necessary. Please remove it from the abstract.
2) Instead, add some background information supporting the reasoning why this study is important in the abstract (before the objective).
R1. Initially, we thank to reviewer for the support and nice overview of our study. In agreement with the reviewer, we have removed the indicated statement from the Abstract. In addition, we have added the background information, as follows (Abstract, page 1, lines 26-28, marked in red): “Recent studies suggested the potential role of mast cells (MCs) in the pathology of coronavirus disease 2019 (COVID-19). However, the precise description of the MCs activation and the engagement of their proteases is still missing.”
Q2. 3) Please specify the severity of COVID-19 included in this study before death: mild, moderate, severe, or critical.
R2. All patients included in the main group had critical COVID-19 and died as a result of it. As suggested by the reviewer, we added this information (page 3, line 101).
Q3. 4) Why didn't the authors check D-dimer in this study? It has been shown in numerous studies that d-dimer is a marker of coagulation and thrombosis in COVID-19 (PMID: 37092518) and high D-dimer is associated with mortality in COVID-19 (PMID: 33395786).
R3. The reviewer is fully right; this indicator should have been taken into account. We'll consider it in the future studies.
Q4. 5) What was expected from procalcitonin measurement in this study? Is it also increased in viral infections such as COVID-19?
R4. An increase in procalcitonin indicates the accession of bacterial infection and correlates with the severity of COVID-19. Thank you very much for your valuable comments. Taking into account the questions that have arisen regarding the correlation analysis of procalcitonin and MCs indicators, to avoid contradictions we decided to exclude this part from the article.
Q5. 6) Line 69: I think the authors need to explain briefly how these proteases are formed and dispatched from MCs. This publication might be helpful (https://doi.org/10.1016/j.mbm.2024.100041).
R5. In agreement with the reviewer, we have described how the MCs proteases are formed and dispatched in the revised manuscript version, as follows (Introduction, page 2, lines 70-74, marked in red): “Briefly, these proteases are synthesized and stored in the cytoplasmic granules of MCs. MCs activation is a complex process initiated by the binding of the IgE antibodies to the MCs receptor FcεRI [8]. This triggers the cascade of intracellular signaling events with the ultimate occurrence of MCs degranulation and release of different active mediators, including the proteases.”
Q6. 7) Please specify the ethical approval number provided by Voronezh State Medical University in line 89.
8) Is there any separate ethical approval from the hospital where the autopsy was done? If so, please also include it.
R6. The Ethics Committee of the Voronezh State Medical University is a responsible authority for the hospital where the autopsy was performed. We have indicated the ethical approval number (page 3, lines 108-109, marked in red).
Q7. 9) The exclusion criteria described in line 94 onward were discovered during autopsy or from medical record when the patients were still alive? Please clarify.
R7. The exclusion criteria were discovered from medical records. We added this information (pages 2-3, lines 88-102, marked in red).
Q8. 10) I think it is very important to provide the flowchart of study design as at the moment, it is quite confusing to understand the step-by-step experimental procedures. For example, which data were taken while the patients were still alive and which were taken after death.
11) How many patients are excluded? due to? Include it in the flowchart
R8. Thank you for your valuable advice! In agreement, we added Figure 1 «Flowchart of study design» with necessary information (page 3).
Q9. 12) Why did the authors use IHC instead of Western Blot or flowcytometry to detect MC proteases? I am afraid that IHC will provide a subjective (operator-dependent) result. Please comment on this important issue and perform additional experiments to confirm the current IHC findings.
R9. We thank to the reviewer for raising up this important issue. Western Blot or flow cytometry are undoubtedly also reliable methods of quantitative analysis of cellular and secreted substances. Changes in different parts of the lung tissue occur unevenly, and then the histological features may differ. We used IHC to evaluate mast cell proteases in order to get an accurate idea of their distribution in the lung regions of interest to us. Importantly, in our work it was important to determine the features of cytotopography, microenvironment, and biogenesis of mast cell proteases in the pathology of COVID-19. We would like to explain that the usage of IHC and subsequent morphometric analysis is a usual method for the MCs research and there are numerous published articles using this approach. Compared to the Western Blot, by using IHC we can obtain information not only about the expression/detection of the particular MCs protease, but also about its localization in the lung tissues. In addition, the quantitative analysis was performed in a blinded-fashion manner, which reduces the subjectivity of the operator-dependent analysis. Although it is not feasible to perform the additional western blot or flow cytometry analysis now, we do agree that the future studies should confirm our finding by using different approaches.
Q10. 13) It is interesting that CVD was not excluded in this study, while it has been shown that MCs are involved in the pathogenesis of CVD including hypertension (PMID: 31295950; 35048969). Why?
R10. It is difficult to recruit a sufficient cohort of patients with COVID-19 without arterial hypertension. The control group is similar to the main group in terms of the presence of comorbidity, including CVD. Thus, differences between groups are explained only by COVID-19 effects.
Q11. 14) in Figure 1, I do think that it is better to also show the histological images of lung tissues from people who died not due to COVID-19 side by side as comparison. Please provide.
15) Also, when explaining Figure 1, please provide arrows or something to show for instance the fibrosis, desquamation, etc. Not all of the readers are pathologists.
R11. 14) Thank you for your valuable advice! We added Figure 1 (revised Figure 2) with recommended examples (non-COVID-19 as comparison) in the revised manuscript (revised Figure 2, page 6, lines 178-190, marked in red).
15) We fully agree with the reviewer, and subsequently we marked the relevant features with the arrows and stars (revised Figure 2, page 6, lines 178-190, marked in red).
Q12. 16) Similarly, add arrows in Figure 2.
R12. We marked the relevant features with the arrows and stars, as recommended by the reviewer (revised Figure 3, page 7, lines 192-201, marked in red).
Q13. 17) Please explain the quantification methods from IHC samples done in Tables 2-4.
R13. Thank you for your question. Histological sections stained immunohistochemically were subjected to a standardized quantitative analysis procedure. The immuno-positive MCs were counted and calculated using a ×40 lens with analysis of at least 50 fields of view. The quantification of the MCs content was performed using counting program incorporated in the AxioVision software. The data is presented per 1 mm2 calculated using the formula 1000000* the total number of cells in 50 fields of view / the total of the area fields of view. The description of the quantification is marked in red in the revised manuscript (page 4, lines 138-143, marked in red).
Q14. 18) Please explain why in Figure 8 we saw procalcitonin only on 0 or 10 ng/ml? Isn't it weird as it was like a qualitative measurement (present or absent)? I don't think it is appropriate to make a correlation if the data is nominal. Also, explain why only tryptase was compared to procalcitonin?
R14. Thank you very much for your advice. To avoid contradictions we excluded this part from the article, as we have already explained by answering to the reviewer´s previous question number 4.
Round 2
Reviewer 1 Report
Comments and Suggestions for Authors
Please cite parallel studies.
Comments on the Quality of English LanguageGood
Author Response
Reviewer 1
Q1. Please cite parallel studies.
R1. We are thankful to the reviewer´s comment. Parallel studies are listed below the references 13,14,15,16,19 (marked in text in yellow).
Reviewer 3 Report
Comments and Suggestions for Authors
Thank you for sending me the response to my previous comments. I only have one minor suggestion:
1) Since the authors decided not to perform additional experiments regarding the d-dimer and western blot in this paper, I suggest to include them as recommendations for future studies, highlighting their importance as already explained in their responses.
Comments on the Quality of English LanguageNo comment
Author Response
Reviewer 3
Q1. Thank you for sending me the response to my previous comments. I only have one minor suggestion: Since the authors decided not to perform additional experiments regarding the d-dimer and western blot in this paper, I suggest to include them as recommendations for future studies, highlighting their importance as already explained in their responses.
R1. Thank you for your suggestion. We included it in lines 424-431: «It is worth noting that in this study we did not perform experiments regarding the D-dimer. Numerous studies showed that D-dimer is a marker of coagulation and thrombosis in COVID-19 and high D-dimer is associated with mortality in COVID-19 [28, 29]. Taking this into account, we consider the possibility of further studies includ-ing D-dimer and recommend it to other authors. Besides, in future studies it is also planned to conduct a western blot to detect MC proteases, which, in addition to immunohistochemical analysis, will ensure maximum accuracy and objectivity of the data.»